# EEG theta and alpha oscillations during tactical decision-making: An examination of the neural efficiency hypothesis in volleyball

Thomas Kanatschnig[1,*], Norbert Schrapf[2], Lisa Leitner[1], Guilherme Wood[1,3], Christof Körner[1,3], Markus Tilp[2,3], Silvia Erika Kober[1,3]

1 Department of Psychology, University of Graz, Graz, Austria, 2 Department of Human Movement Science, Sport and Health, University of Graz, Graz, Austria, 3 BioTechMed-Graz, Graz, Austria

* thomas.kanatschnig@uni-graz.at

**Data availability statement:** All analysis data and code used for statistical calculations and complementary analyses, either presented or discussed in this article, are available at the OSF project page of this study: https://osf.io/ck5zu/

**Funding:** This study was financially supported by University of Graz for coverage of

## Abstract

According to the neural efficiency hypothesis (NEH), individuals with higher expertise in a domain use their brain more efficiently when processing domain-specific tasks and show more efficient brain activity than individuals with lower expertise. In this study 64 participants with differing levels of volleyball expertise were recruited to investigate the NEH by means of a volleyball-specific tactical decision-making task. The participants, which were allocated to three different expertise groups (based on prior volleyball experience), saw videos of setting situations of real volleyball matches and were asked to predict the outcome of these situations. Behavioral performance and event-related de-/synchronization (ERD/S) in the EEG theta and alpha bands during the completion of the task, were examined. Our results show significantly higher prediction accuracy in relation to volleyball expertise. Significantly stronger theta synchronization, as well as alpha desynchronization were observed during the tactical prediction condition compared to a control condition. No significant group differences in theta or alpha ERD/S were observed. Descriptive examinations of theta and alpha ERD/S distributions, which revealed mixed results regarding support for the NEH, are discussed. Our findings provide new insights to the ongoing debate on how the NEH applies to the domain of sport.

## Introduction

### Optimal sport performance

Volleyball is one of the most popular sports in the world. As in any other sport, there is great interest from professional volleyball players as well as coaches to maximize competitive performance. A multitude of factors play a role for athletic success, such as genetic disposition [1], visuo-perceptual functioning [2] or team cohesion [3]. Yet, especially in the case of team-based sports, studying ways to improve a team's performance poses a challenge to researchers, given the many degrees of freedom that arise when multiple athletes share the same court. To understand the complex interplay between athletes in team-based sports, strategy and tactical assessment play an important role as well [4,5]. In their review on decision-making in youth team-sports, Silva et al. [5] conclude that higher-level as well as older athletes have

publication fees. No additional external funding was received for this study.

**Competing interests:** The authors have declared that no competing interests exist.

an advantage against their lower-level and younger counterparts when it comes to resolving complex tactical situations. This suggests that already at a young age the amount of expertise (i.e., training and experience) an athlete accumulates improves their tactical skill, which in turn can give them the necessary advantage in competition. As Araújo et al. [6] summarize, decision-making in sports is a multi-layered dynamic process; action modes emerge from given affordances, which are adjusted constantly based on the continuous stream of information an athlete receives.

## The neural efficiency hypothesis

There have been numerous investigations in the field of neuroscience to understand how expertise is manifested on a neurophysiological level. Haier et al. [7] were the first to describe what was later to become known as the neural efficiency hypothesis (NEH). They found a negative correlation between measures of intelligence and the cerebral glucose metabolic rate as an indicator for brain activation, measured through positron emission tomography. This finding was the basis for the formation of the NEH, that in brief states that more intelligent individuals exhibit lower (i.e., more efficient) brain activation during cognitive tasks compared to less intelligent individuals [8]. While many studies using other neuroimaging techniques, such as electroencephalography (EEG) and functional magnetic resonance imaging (fMRI), came to similar results as Haier et al. [7], some also reported conflicting results or discovered moderating factors for the relationship between cognitive ability and brain activation (for an overview see Neubauer and Fink [9]). The works by Grabner et al. [10,11] for instance demonstrate support for the NEH in that they could repeatedly show a negative relationship between IQ and brain activation, in the form of task-related changes in EEG alpha band power (i.e., event-related de-/synchronization, ERD/S [12]). Higher IQ was associated with weaker event-related desynchronization (ERD) of the alpha band during an intelligence related performance task, which can be interpreted as more efficient brain functioning by more intelligent individuals, as task-related desynchronization of the alpha band is an indicator of cognitive strain [13,14]. Higher alpha ERD has generally been interpreted as stronger cognitive strain in the context of neural efficiency, meaning weaker alpha ERD suggests relatively lower cognitive strain. Accordingly, an event-related synchronization (ERS) of the alpha band, meaning a relative increase in task-related alpha power indicates even less cognitive strain and a more relaxed cognitive state. Conversely, when testing domain-specific performance of taxi drivers in a spatial orientation task, Grabner et al. [11] found the opposite effect, namely a positive relationship of IQ and alpha ERD. Furthermore, the findings from an investigation of working memory performance suggested, that the NEH-effect is connected to fluid but not crystallized intelligence [15].

Although research on the NEH is rooted in intelligence, it later expanded to other domains. This is also true for the domain of sport, where the NEH and cognitive functions in general have been studied with different neuroimaging techniques and across various disciplines, ranging from self-paced sports such as golf, archery, and darts, to team-based sports, such as soccer, basketball, and volleyball [16–18]. In their review, Li and Smith [18] summarize the current state on neuroscientific research on the NEH in sport. They conclude that in expert-novice comparisons, which is the most frequently found procedure for investigations into the NEH in sports, experts generally tend to perform at faster speeds, more accurately, and with greater (neural) efficiency than novices. Yet the authors also highlight that not all studies they reviewed found support for the NEH, some of which also show contradictory results, indicating not weaker but stronger brain activation of experts compared to novices. They name this circumstance the "efficiency paradox" and discuss neural proficiency as a complementary aspect to the NEH. In their review on neural efficiency in self-paced sports,

Filho et al. [17] summarize patterns of brain activity specific to the EEG theta and alpha bands. From the literature they derived a trend, that experts show relatively lower theta and higher alpha power compared to novices during a sport-specific task. Increased task-related theta power, i.e., ERS of theta, has been associated with higher mental activity [19–21], whereas increased task-related alpha power, i.e., ERS of alpha, is associated with processes of cognitive inhibition and relaxation [12,14,22]. Accordingly, more efficient brain functioning during task performance can be described as relatively weaker theta ERS (stronger theta ERD) and relatively stronger alpha ERS (weaker alpha ERD), which indicates lower mental activity. Especially regarding the alpha band, less ERD in experts compared to novices is a recurring pattern in the literature. Findings by Babiloni and colleagues showed weaker alpha ERD in relation to expertise level of rhythmic gymnasts [23], as well as karate athletes [24], during domain-specific cognitive tasks. Similarly, findings by Del Percio and colleagues showed weaker alpha ERD in experts compared to novices during pistol shooting [25], a standardized balancing task [26], and voluntary hand movements [27]. More recently however, Del Percio et al. [28] investigated the brain activity of soccer players during a visuospatial soccer task and found stronger alpha ERD in experts compared to novices, thereby contradicting the NEH.

## Neural efficiency in volleyball

Out of the 28 studies Li and Smith [18] selected in their review, only one focused on volleyball. In that study, Zhang et al. [29] examined the relationship between the brain activity of volleyball and basketball players during the visualization of sport-specific movements by means of fMRI. They found lower brain activation during imagination of a "self-sport" movement compared to an "other-sport" movement, which indicates more efficient brain activity of athletes in relation to their own sport compared to a non-familiar sport. In another recent study, DeCouto et al. [30] examined behavioral as well as neurophysiological differences between more and less skilled volleyball players during an attentional priming paradigm using temporally occluded videos of volleyball attacks, for which the outcome had to be predicted. They found generally higher accuracy in performance of skilled players compared to less skilled players, as well as significantly lower absolute EEG alpha and beta power in the right compared to the left parietal region but only in the skilled group of players. The authors described their findings as an indication of right parietal dominance for skilled attack anticipation in volleyball players. Taken together, neuroscientific investigations of the game of volleyball exist, albeit more concrete insight into the role of the NEH is lacking.

## Present study

To summarize, although there is manifold research on the NEH in sport, investigations into the domain of volleyball are scarce. Moreover, we did not find neuroscientific studies on volleyball using the EEG-ERD/S approach, which is an established method for investigating task-related changes in brain activity in NEH research. Also, patterns of brain activation observed in different studies do not reliably support the NEH, which demonstrates the need for more comprehensive study designs. With the present study, we present to our knowledge the first investigation of the NEH in tactical volleyball expertise. We conducted an experiment, in which we measured the performance and brain activation of participants during the completion of a volleyball-specific tactical decision-making task, for which we used the stimulus material of a previous study. In that study, Schrapf et al. [31] gave volleyball coaches an earlier version of the tactical decision-making task, in which videos of volleyball setting situations were presented. In the present study, an adapted version of this tactical decision-making task was presented to participants. Analogously to previous studies

(e.g., [24]), we recruited three groups of participants with different levels of prior volleyball experience. We used EEG to measure task-related changes in brain activation, i.e., ERD/S. Based on previous findings regarding the NEH in sports, we expected to find indications of neural efficiency. Our main EEG frequency bands of interest were theta and alpha. Given the current state of research on neural efficiency, it was hypothesized that individuals with higher expertise in the domain of volleyball exhibit more efficient brain activation (i.e., weaker theta ERS, weaker alpha ERD) during the completion of a volleyball-specific tactical decision-making task, relative to individuals with lower volleyball expertise. Furthermore, we expected to find better behavioral performance (i.e., higher prediction accuracy, shorter response time) in individuals with higher expertise compared to individuals with lower expertise.

## Method

### Participants

A total of 64 right-handed participants took part in this study. Among them were 38 women and 26 men, ranging in age from 18 to 42 years ($M = 23.27$ years, $SD = 4.47$). Two participants stated that they had completed a doctorate or vocational college. The remaining 62 participants were studying for a bachelor's or master's degree at the time of the study. To investigate volleyball expertise on a broad spectrum, three groups of participants with different levels of volleyball experience were recruited. The following operational criteria were used to allocate participants to one of the three expertise groups. The *Expert* group ($n = 16$; 11 women; $M = 23.10$ years, $SD = 5.89$) consisted of professional volleyball players who actively played in the first or second national volleyball league of Austria. One participant of the Expert group was not a player anymore at the time of data collection, but a professional national league coach. The experts were the group with the highest volleyball expertise. Next, the *Amateur* group ($n = 26$; 13 women; $M = 22.80$ years, $SD = 4.51$) consisted of individuals who did not play professionally but had a good understanding of volleyball. They were actively playing in a volleyball club or in a university volleyball course at intermediate or advanced level. The amateurs therefore had a medium level of volleyball expertise. Lastly, the *Novice* group ($n = 22$; 14 women; $M = 24.00$ years, $SD = 3.20$) included individuals who had little to no previous volleyball experience, as well as ones who may have played in a volleyball club in previous years. The novices were the group with the lowest volleyball expertise. All participants received detailed information about the procedures before the start of the study and signed a written consent form. Participants received a compensation of 24€ for their participation. Psychology students could instead gather course credit. All procedures of the present study have been conducted in accordance with the Declaration of Helsinki and were approved by the ethics committee of the University of Graz (GZ. 39/105/63 ex 2021/22). Recruitment and data collection was performed from September 1, 2022 until February 28, 2023, via contact to local volleyball clubs, as well as through advertisement on social media platforms, e-mail distribution and flyer postings. The procedures for this study were preregistered at the Open Science Framework (OSF; https://osf.io/exhr3; deviations from the preregistered methods are listed in Table A in S1 Tables under "Supporting information").

### Materials

**Tactical decision-making task.** The tactical decision-making task for volleyball, which is an adaptation of the task used previously in our research with volleyball coaches [31], consisted of in total 63 short video stimuli which all showed a common game situation in 6 versus 6 women's indoor volleyball and in which the main subject of interest, the final

pass location of the setter player's pass, was temporally occluded, meaning the videos ended exactly when the setter received the ball and before the pass was played. Before each video, participants were asked to fixate their gaze for 2 s on a fixation disk in the middle of a screen showing a grey background; this was the *Baseline* phase. After that, the video for the trial started. The videos ranged between 3 and 13 s in length (depending on the amount of time players took to prepare the rally in each respective video). Every video started with showing the service player of one team preparing for the service and ultimately playing the service to start the rally. The opposing team then received the ball and prepared their attack, by passing the ball to a tactically advantageous position. Every video stopped when the setter player of the attacking team received the ball. This player was responsible for playing the final pass, also referred to as the "setting", to a teammate for the attack. The last 3 s of each video, which was the time interval in which the main action of the rally happens (i.e., the time between the service and the point where the setter received the ball) were defined as the *Rally* phase. The task consisted of two conditions. In the *Prediction* condition, participants viewed each video and were asked to then make a prediction to which position the setter player will pass the ball ("Where will the SETTER pass the ball to?"). In the *Control* condition, participants viewed the same videos a second time, without the task to predict the setter's pass, but to simply name the position of the service player during the service ("Where does the SERVICE player stand when serving?"). At the end of the video, the last frame (immediately before the pass of the setter) was displayed to the participants for 0.5 s as a freeze frame, which was defined as the *Freeze* phase. Following that, the response screen appeared, showing the outlines of a volleyball court with the respective answer options for the Prediction condition (4 possible pass destination locations of the setter player) and Control condition (3 possible service locations of the service player) and participants were asked to give their response; this was the *Response* phase. Responses should be given not only well-deliberated (based on the course of the rally) but also as fast as possible via button press. There was no time limit for participants' responses. Furthermore, the task was divided into two separate blocks. In the *Near-side* block the actions of interest (i.e., the setting in the Prediction condition and the service in the Control condition) occurred on the nearer side of the court (court side in front of the net), whereas in the *Far-side* block the actions occurred on the farther side of the court (court side behind the net), with respect to the camera's field of view of the video recordings. The number of videos for the Prediction condition was 25 in the Near-side and 38 in the Far-side block. The videos of the Prediction condition in one block functioned as videos for the Control condition in the other block, given that the service always happened on the opposite court side. Videos were presented without sound and in randomized order. For a visualization of the tactical decision-making task see Fig 1.

**Questionnaires.** A short questionnaire to collect sociodemographic data (e.g., age, sex, occupation, etc.) was presented to the participants. Among these questionnaire items, there were items about prior volleyball experience and sport habits, a part of which were used to evaluate our criteria for the group classification of participants. The Edinburgh Handedness Inventory (EHI; Oldfield [32]) was used to assess handedness. The EHI is a short questionnaire that records a person's hand dominance using 10 items. These items describe everyday activities in which it must be stated which hand is used for this activity (left, right or both) and whether the other hand is ever used for this activity. The resulting laterality index can range between -100 and +100. High negative values are associated with left-handedness, values around 0 with ambidexterity, and high positive values with right-handedness.

Based on the connection between high-level performance and flow experience suggested in previous research [17,33,34] we presented the Flow Short Scale (original title: Flow-Kurzskala;

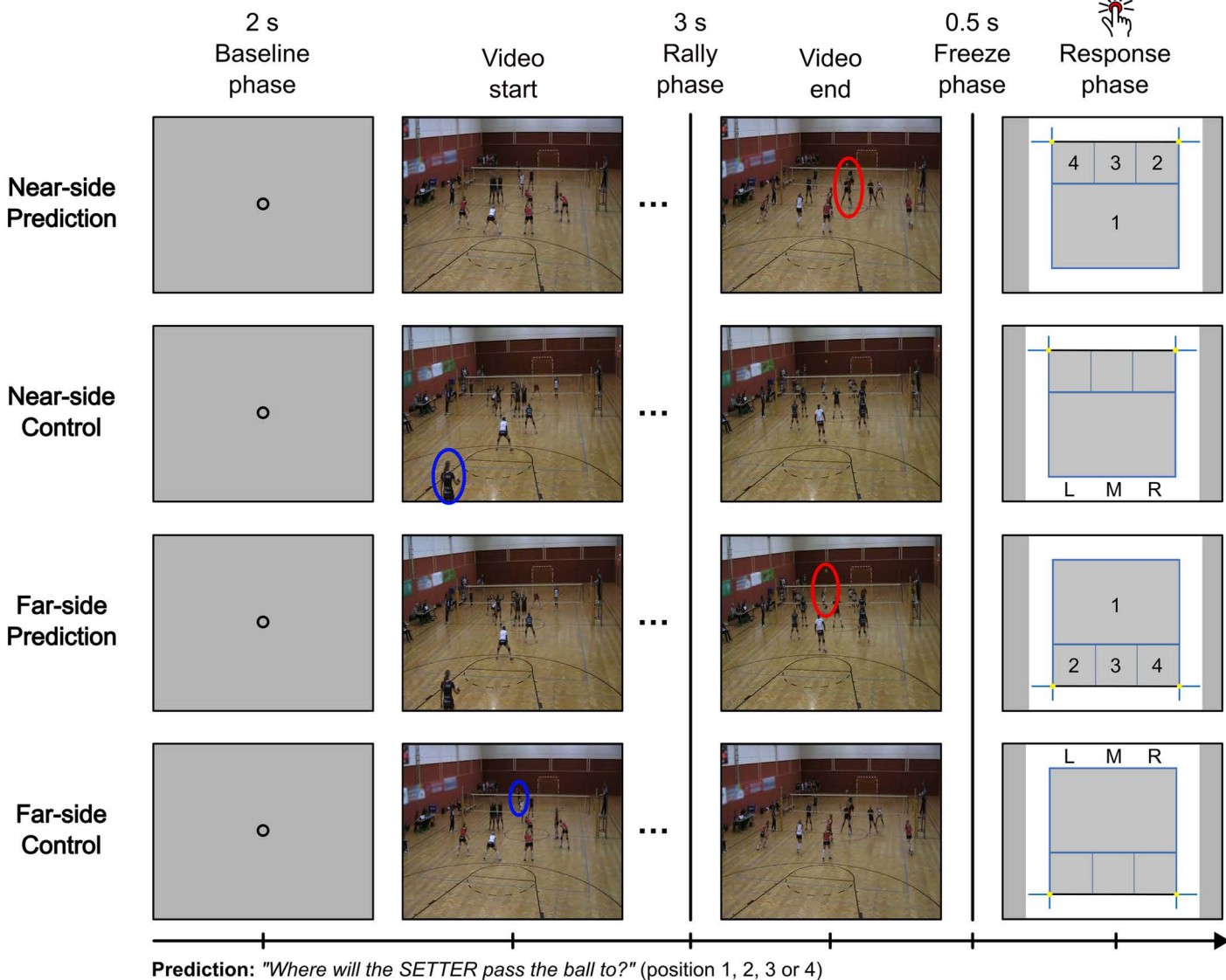

**Fig 1. Visualization of the tactical decision-making task.** The task was conducted in two separate blocks, Near-side and Far-side. Each block included a Prediction and a Control condition, the order of which was counterbalanced across participants in both blocks. Each trial consisted of a 2 s Baseline phase for gaze fixation after which the video started. The videos varied in length. The last 3 s of each video is referred to as the Rally phase, which is the time interval in which the main action of the rally happened in the videos. When the video ended, the last frame of the video remained visible for 0.5 s in the Freeze phase. Immediately after that, the response screen appeared, and participants were asked to give their response in the Response phase. The locations of the respective players of interest, namely the setter in the Prediction condition and the service player in the Control condition, are highlighted in red and blue, respectively.

FKS; Rheinberg et al. [35]) questionnaire on flow experience to our participants. The FKS consists of 16 items that can be divided into 3 parts: (i) the core 10 item *Flow Short Scale* (e.g., "I'm completely absorbed in what I'm doing right now."), (ii) a 3-item extension measuring *Concerns of Failure* (e.g., "I can't make any mistakes now.") and (iii) another 3-item extension measuring subjective *Feeling of Ability* to complete the given task (e.g., "I think my skills in this area are... [low/high]"). Items of parts (i) and (ii) are each scored on a 7-point Likert scale ranging from 1 ("does not apply") to 7 ("does apply"), where higher values indicate higher

flow experience and higher concern, respectively. Items of part (iii) vary in their scaling type which makes it difficult to derive a composite score out of it. Since we could not find conclusive information on how to interpret the items concerning the ability scale, we decided to not include them in the analysis.

Additionally, visual analogue scale (VAS) items were used to record the subjective assessment of task difficulty, motivation, concentration, mood, as well as task performance of the participants during the tactical decision-making task. By positioning a slider on a horizontal line, participants gave their assessment on the respective measures (the analysis of the VAS items is presented in Table C in S1 Tables).

## Design and general procedure

Our experiment included the following independent variables: (i) *Group*, i.e., Novice vs. Amateur vs. Expert group, (ii) *Court*, i.e., Near-side vs. Far-side, and (iii) *Task*, i.e., Prediction vs. Control. Although we did not formulate hypotheses concerning the variable Court, we wanted to control its effect, since we assumed that the different perspectives on the actions of interest in the videos could potentially influence participants' performance. This is an extension in regard to our previous work [31] as this aspect of the stimulus material has not yet been investigated. The main dependent variables of our investigation can be categorized into (i) behavioral performance variables, and (ii) neurophysiological variables. In terms of behavioral performance, we measured accuracy (i.e., the percentage of correct responses) and response time (i.e., the average time between appearance of the response screen and the response of the participant). Regarding neurophysiological variables, we specifically focused on EEG theta and alpha. We particularly looked at task-related changes in theta and alpha power, i.e., ERD/S of the theta and alpha bands, at different time points, i.e., the Rally and Freeze phases of the tactical decision-making task, in relation to the Baseline phase.

Data acquisition took place in a sound attenuated and electrically shielded EEG laboratory. After participants filled out the first part of questionnaires (sociodemographic variables, EHI) via LimeSurvey (LimeSurvey GmbH [36]), their head was positioned on a chin rest so that it was at a fixed distance of 62 cm from the computer screen (24.4-inch LCD display). During the experiment we additionally tracked the eye movements of participants, to which however, we will not go into further detail in the present work, as those data were treated separately and not in combination with the EEG data. After the EEG montage was completed, two 2-minute resting measurements were recorded, one with eyes open and one with eyes closed. Then, the main part began, i.e., the tactical decision-making task (see section "Tactical decision-making task"). Before the first task block, participants carried out a practice run with different stimuli to get familiar with the task. Participants completed the first block, which was the Near-side block, in which the order of Prediction and Control conditions was counterbalanced across participants. After completing the Near-side block, participants carried out another training run for the second block, which was the Far-side block. During the Far-side block the order of Prediction and Control conditions was again counterbalanced. The implementation of the tactical decision-making task was carried out with PsychoPy (version 2021.2.3, Peirce et al. [37]). Due to drift correction for the eye tracking, the inter-trial interval (ITI) was not consistent. The ITI consisted of a randomized 3–4 s pause showing a blank screen, followed by the appearance of the fixation disk in the middle of the screen for drift correction (1–2 s for each trial). After completing the tactical decision-making task, the second part of the LimeSurvey was performed (flow experience, VAS), in which the participants were also asked to give written responses about which type of strategies they used to perform the task.

## EEG recording and pre-processing

A total of 55 scalp EEG channels were measured, with the electrodes placed according to the international 10–20 system. Given that a chin rest was used that also included a support bar for the forehead (a necessary means to stabilize the head for the eye tracking), some of the frontal channels usually present in a comparable EEG montage had to be omitted. Two reference electrodes were placed on the right and left mastoids behind the ears, respectively. The ground electrode (GND) was positioned at AFZ. To record horizontal and vertical eye movements, three EOG electrodes were used, which were positioned based on the placement scheme by Schlögl et al. [38]. Care was taken to keep the impedances of the ground electrode and the reference electrodes below 10 kΩ and the impedances of the scalp electrodes below 25 kΩ. We used the actiCAP slim active electrode system together with BrainAmp amplifiers from Brain Products GmbH (Gilching, Germany). The EEG was recorded using the BrainVision Recorder software (version 2.2, BrainProducts GmbH, Munich, Germany) with a sampling rate of 500 Hz.

The recorded EEG data was pre-processed using the BrainVision Analyzer software (version 2.2, BrainProducts GmbH, Munich, Germany). First, a new reference (linked-mastoid reference) was formed from the two reference channels. Manual artifact correction was then carried out, in which larger muscle artifacts were cut. To eliminate power line interference a 50 Hz notch filter was used. Furthermore, artifacts caused by eye movements were identified and corrected using ICA. This was followed by a semi-automatic artifact identification with the following EEG criteria: (i) > 50 µV voltage difference between two data points, (ii) > 200 µV voltage difference within a 200 ms interval and (iii) absolute voltage values ±200 µV. EEG signal that fell under one of those criteria was marked as bad 500 ms before and after the identified datapoint. Remaining artifacts not detected by these criteria were manually cut and EEG channels that had noisy signal were excluded from further analysis. For some data sets it was necessary to adjust the standard artifact correction criteria. The EEG data of four participants (participant codes 13, 32, 52 and 60) had to be excluded completely from further analysis due to poor signal quality or incomplete data. Following artifact correction, extraction of theta (4–7 Hz) and alpha (8–12 Hz) power was performed using the "Complex Demodulation" transformation implemented in the BrainVision Analyzer software. Segmentation and averaging of theta and alpha power data was performed for the Baseline, Rally, and Freeze phases, respectively. Averaged power data for theta and alpha was then further processed using R (version 4.2.2, R Core Team [39]), RStudio (version 2023.06.0 + 421, RStudio Team [40]) and the *tidyverse* package (version 2.0.0, Wickham et al. [41]). Event-related de-/synchronization (ERD/S) values were calculated for theta and alpha, for each condition (Prediction, Control), block (Near-side, Far-side), and time interval of interest (Rally, Freeze) using the formula:

$$ERD(S)\% = \frac{(A - R)}{R} * 100 \tag{1}$$

In this formula "A" stands for the active time period, i.e., Rally or Freeze, and "R" stands for the resting time period, i.e., Baseline [12]. Positive values resulting from this formula therefore signify an event-related synchronization (ERS), while negative values signify an event-related desynchronization (ERD).

For the comparison of task-related theta and alpha band activity between groups and conditions average ERD/S of the frontal midline region (FZ, FCZ, CZ) was used for theta, while average ERD/S of the parietal region (P5, P3, P1, PZ, P2, P4, P6) was used for alpha. Our considerations for the choice of these specific regions of interest and the electrode selection were based on previous research. Capilla et al. [42] demonstrated that resting activity in the frontal midline area is especially linked to the theta band, whereas in the parietal area it is linked to

the alpha band. Also during active cognition, frontal midline regions have shown to be highly sensitive to theta, while parietal regions are to alpha [19–21,23,24,43]. See Fig 2 for a visualization of the EEG montage and regions of interest that were examined in this study.

## Statistical analysis

Questionnaire data were examined using $\chi^2$ tests and Cramer's *V* calculations in the case of categorical variables, as well as between-subjects ANOVAs in the case of continuous variables. Behavioral performance differences in task accuracy and response time were tested using two separate 2x2x3 mixed-design ANOVAs, which included the between-subjects factor Group (Novice, Amateur, Expert), as well as the within-subjects factors Court (Near-side, Far-side) and Task (Prediction, Control). To tackle the issue of strong deviations from the normal distribution in the case of response time, a log-transformation was performed. Log-transformed response times were used for statistical calculations; however, for interpretability, non-log-transformed response times are presented for descriptive statistics and visualization. On the neurophysiological level, frontal

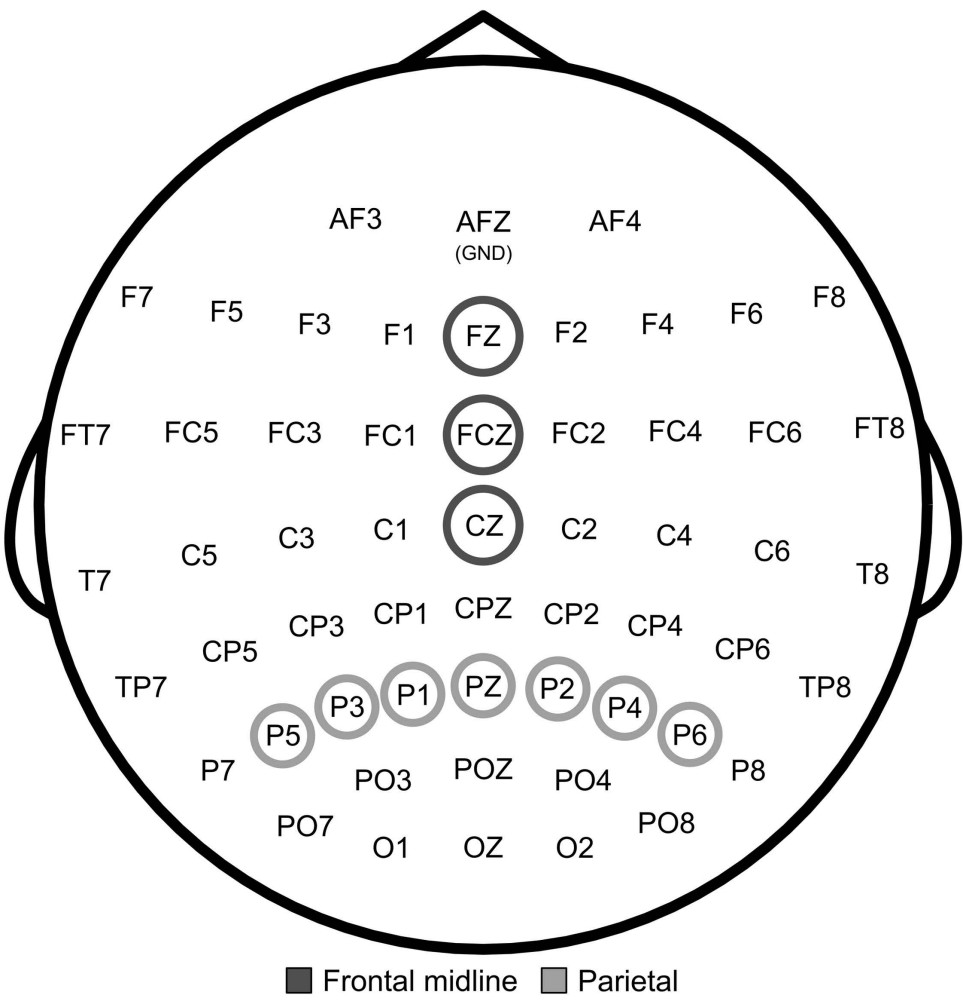

**Fig 2. Schematic representation of the EEG montage.** The electrode positioning was based on the international 10–20 system. The GND electrode was positioned at AFZ. Marked EEG channels indicate the examined regions of interest (dark grey: Frontal midline area; light grey: Parietal area).

midline theta and parietal alpha ERD/S values were tested each with 2 separate 2x2x3 mixed-design ANOVAs (including also the factors Group, Court, and Task) for the Rally phase (last 3 s before the end of each video) and Freeze phase (0.5 s freeze of the last video frame), respectively.

Statistical calculations were performed with base R functions and the *rstatix* package (version 0.7.2., Kassambara [44]). The packages *ggpubr* (version 0.6.0, Kassambara [45]) and *eegUtils* (version 0.7.0, Craddock [46]) were used for visualizations. The significance level for all analyses was set to $\alpha$ = 0.05 (two-tailed). Post-hoc comparisons were performed with *t*-tests, for which (in the case of independent samples comparisons) the degrees of freedom were adjusted by means of the Welch method [47]. When analyzing multiple pairwise comparisons, the significance level was automatically adjusted using the Holm method [48]. Prerequisites for ANOVA calculations, namely normal distribution and homoscedasticity were examined using Shapiro-Wilk and Levene's tests, respectively. While the criterion of homoscedasticity was met in almost all above-described statistical calculations (the only exception being the analysis of response time), deviations from normal distribution were present in multiple cases, albeit not to a great extent. Prerequisite statistics will not be discussed further but can be examined alongside all main as well as complementary analyses using the provided online materials, which include all data and analysis scripts used for this article [49].

Given the fact that interindividual differences in EEG frequency band limits have been deemed an important aspect to consider for the analysis of ERD/S data [19], we conducted complementary analyses to show that the three groups did not differ significantly in individual alpha frequency (see Table D in S1 Tables). Given these results, we proceeded with fixed frequency band limits for alpha (8–12 Hz) as well as theta (4–7 Hz).

## Results

### Questionnaire variables analysis

Table 1 shows comparative statistics for the main questionnaire variables of interest. The variables presented here are what we deemed to be the most informative ones (Table B in S1 Tables gives a detailed description of all questionnaire variables). We were aiming for a balanced distribution in terms of sex, age, and EHI score across groups. Furthermore, we expected prior volleyball experience to differ between the three expertise groups. Groups did not differ in terms of sex, age and EHI. Tests for all variables regarding prior volleyball experience yielded a significant result, indicating differences between the three groups. Furthermore, when asked whether they performed other sports besides volleyball regularly, most of the participants responded "yes" and there was no difference found in the distribution across groups, indicating that in general the sample consisted of active individuals.

Flow experience during the tactical decision-making task was examined using the Flow Short Scale and the extension scale measuring Concerns of Failure [35]. A significant effect was found for the Flow Short Scale. Post-hoc comparisons revealed that the experts reported higher flow experience compared to the amateurs. No difference was found between novices and experts and between novices and amateurs. No group differences were found on the Concern of Failure scale.

### Behavioral task performance analysis

The ANOVA for task accuracy (percentage of correct responses) yielded significant main effects for the factors Group and Task, as well as significant interaction effects for Group × Task and Court × Task. No other main or interaction effects were significant. Post-hoc comparisons for the interaction effect Group × Task revealed that in the Prediction condition experts ($M$ = 67.0, $SD$ = 7.57) reached significantly higher accuracy compared to amateurs

**Table 1. Comparative statistics for main questionnaire variables.** Questionnaire variable labels are presented in the "Variable" column; in the case of categorical variables with the respective categories in parentheses. Variable names are shown in subscript square brackets (for reference see Table B in S1 Tables). Descriptive statistics are presented in the "Group" column. For categorical variables absolute participant counts are presented for the respective categories. For continuous variables group means are presented, with standard deviations in parentheses. Comparative statistics results (categorical: $\chi 2$ tests + Cramer's $V$ calculations; continuous: between-subjects ANOVAs) are provided in the "Statistics" column. Variables with significant overall test results are indicated with asterisks (**: $p < .01$; ***: $p < .001$). For continuous variables, significant group differences, as determined through pairwise $t$-tests, are indicated with uppercase letters ([a-c]).

| Variable: | Group: | | | Statistics: | | | |
|---|---|---|---|---|---|---|---|
| | Novice (n = 22) | Amateur (n = 26) | Expert (n = 16) | df | $\chi 2$ | p | V |
| Sex (female/male) [SOC04] | 14/8 | 13/13 | 11/5 | 2 | 1.70 | .428 | .16 |
| "Do you currently play actively in a volleyball club?" (no/yes) *** [SOC06] | 22/0 | 15/11 | 0/16 | 2 | 38.00 | <.001 | .77 |
| "Do you play volleyball in your free time?" (no/yes) [SOC08] *** | 10/12 | 0/26 | 0/16 | 2 | 22.6 | <.001 | .59 |
| "Are you currently attending one or more volleyball courses at the University Sports Institute (USI)?" (no/yes) [SOC09] *** | 22/0 | 10/16 | 16/0 | 2 | 31.2 | <.001 | .70 |
| "Apart from volleyball, do you play sports regularly?" (no/yes) [SOC16] | 4/18 | 8/18 | 1/15 | 2 | 3.77 | .152 | .24 |
| | | | | df | F | p | $\eta_p^2$ |
| Age in years [SOC03] | 24.00 (±3.20) | 22.80 (±4.51) | 23.10 (±5.89) | 2, 61 | 0.42 | .658 | .014 |
| "On average, how many hours a week do you play volleyball? (Please provide an approximate estimate of the number of hours.)" [SOC12] *** | 0.48$^{ab}$ (±0.79) | 3.79$^{ac}$ (±2.09) | 9.62$^{bc}$ (±2.90) | 2, 61 | 95.66 | <.001 | .758 |
| "Approximately how many years of volleyball experience do you have? (Please provide a rough estimate of the years.)" [SOC13] *** | 4.30$^a$ (±6.39) | 5.73$^b$ (±3.41) | 11.20$^{ab}$ (±3.54) | 2, 61 | 10.79 | <.001 | .261 |
| "On average, how many hours per week do you exercise? (Please provide an approximate estimate of the number of hours.)" [SOC18] *** | 4.18$^a$ (±2.22) | 5.23$^b$ (±2.50) | 9.28$^{ab}$ (±6.04) | 2, 61 | 9.79 | <.001 | .243 |
| Edinburgh Handedness Inventory [EHI] | 75.90 (±18.60) | 78.0 (±19.20) | 74.40 (±16.70) | 2, 61 | 0.29 | .746 | .010 |
| Flow Short Scale [FLOW_S] ** | 4.93 (±0.71) | 4.43$^a$ (±0.87) | 5.26$^a$ (±0.88) | 2, 61 | 5.38 | .007 | .150 |
| Flow Concern Scale [FLOW_C] | 2.74 (±1.70) | 2.92 (±1.50) | 2.06 (±1.52) | 2, 61 | 1.54 | .223 | .048 |

($M = 45.1$, $SD = 9.45$, $t(37.1) = -8.26$, $p < .001$) and novices ($M = 35.8$, $SD = 10.4$, $t(36.0) = -10.7$, $p < .001$), and amateurs also reached higher accuracy than novices ($t(42.9) = -3.22$, $p = .002$). In contrast, in the Control condition there were no significant differences in accuracy between novices ($M = 84.4$, $SD = 18.9$) and amateurs ($M = 90.3$, $SD = 6.8$, $t(25.6) = -1.40$, $p = .522$), novices and experts ($M = 90.0$, $SD = 7.86$, $t(29.8) = -1.25$, $p = .522$), and amateurs and experts ($t(28.4) = 0.13$, $p = .898$). Furthermore, all groups reached significantly higher accuracy in the Control compared to the Prediction condition (Novice: $t(21) = -12.9$, $p < .001$; Amateur: $t(25) = -20.2$, $p < .001$; Expert: $t(15) = -10.8$, $p < .001$). Post-hoc comparisons for the interaction effect Court × Task revealed that average accuracy across all participants was higher in the Far-side ($M = 50.0$, $SD = 15.9$) compared to the Near-side ($M = 44.8$, $SD = 16.1$) block in the Prediction condition ($t(63) = -4.32$, $p < .001$). There was no difference in accuracy between Near-side ($M = 88.8$, $SD = 14.4$) and Far-side ($M = 87.6$, $SD = 13.7$) in the Control condition ($t(63) = 0.82$, $p = .416$). Also, accuracy was higher in the Control compared to the Prediction condition in the Near-side ($t(63) = -19.7$, $p < .001$) as well as in the Far-side block ($t(63) = -15.6$, $p < .001$). See Fig 3 for a visualization of the accuracy results.

The ANOVA for task response time (average log-transformed seconds until response) yielded significant main effects for the factors Court and Task, as well as significant interaction effects for Group × Task and Court × Task. No other main or interaction effects were

significant. Post-hoc comparisons for the interaction effect Group × Task revealed that across all groups, participants were faster to give their response in the Control (Novice: $M = 0.74$, $SD = 0.38$; Amateur: $M = 0.62$, $SD = 0.15$; Expert: $M = 0.52$, $SD = 0.13$) compared to the Prediction condition (Novice: $M = 1.71$, $SD = 0.99$, $t(21) = 11.5$, $p < .001$; Amateur: $M = 1.80$, $SD = 0.43$, $t(25) = 21.7$, $p < .001$; Expert: $M = 1.55$, $SD = 0.39$, $t(15) = 12.8$, $p < .001$). In terms of group differences, no comparison reached significance either in the Prediction (Novice vs. Amateur: $t(28.9) = -1.25$, $p = .442$; Novice vs. Expert: $t(33.1) = 0.10$, $p = .923$; Amateur vs. Expert: $t(29.6) = 1.97$, $p = .174$) or in the Control (Novice vs. Amateur: $t(31.5) = 0.87$, $p = .391$; Novice vs. Expert: $t(33.0) = 2.40$, $p = .067$; Amateur vs. Expert: $t(33.6) = 2.31$, $p = .067$) condition. Post-hoc comparisons for the interaction effect Court × Task revealed that participants' responses were faster during the Far-side compared to the Near-side block, both in the Prediction (Near-side: $M = 1.90$, $SD = 0.71$; Far-side: $M = 1.51$, $SD = 0.68$, $t(63) = 9.36$, $p < .001$) as well as in the Control condition (Near-side: $M = 0.70$, $SD = 0.30$; Far-side: $M = 0.58$, $SD = 0.25$, $t(63) = 5.92$, $p < .001$). Also, response times were faster in the Control compared to the Prediction condition, in the Near-side ($t(63) = 23.9$, $p < .001$) as well as the Far-side block ($t(63) = 21.2$, $p < .001$). See Fig 4 for a visualization of the response time results. Complete ANOVA results for accuracy and response time analyses are presented in Table 2.

### Theta ERD/S analysis

The analyses of theta ERD/S during the tactical decision-making task are presented separately for the Rally and Freeze phases. Positive values indicate an ERS, negative values indicate an

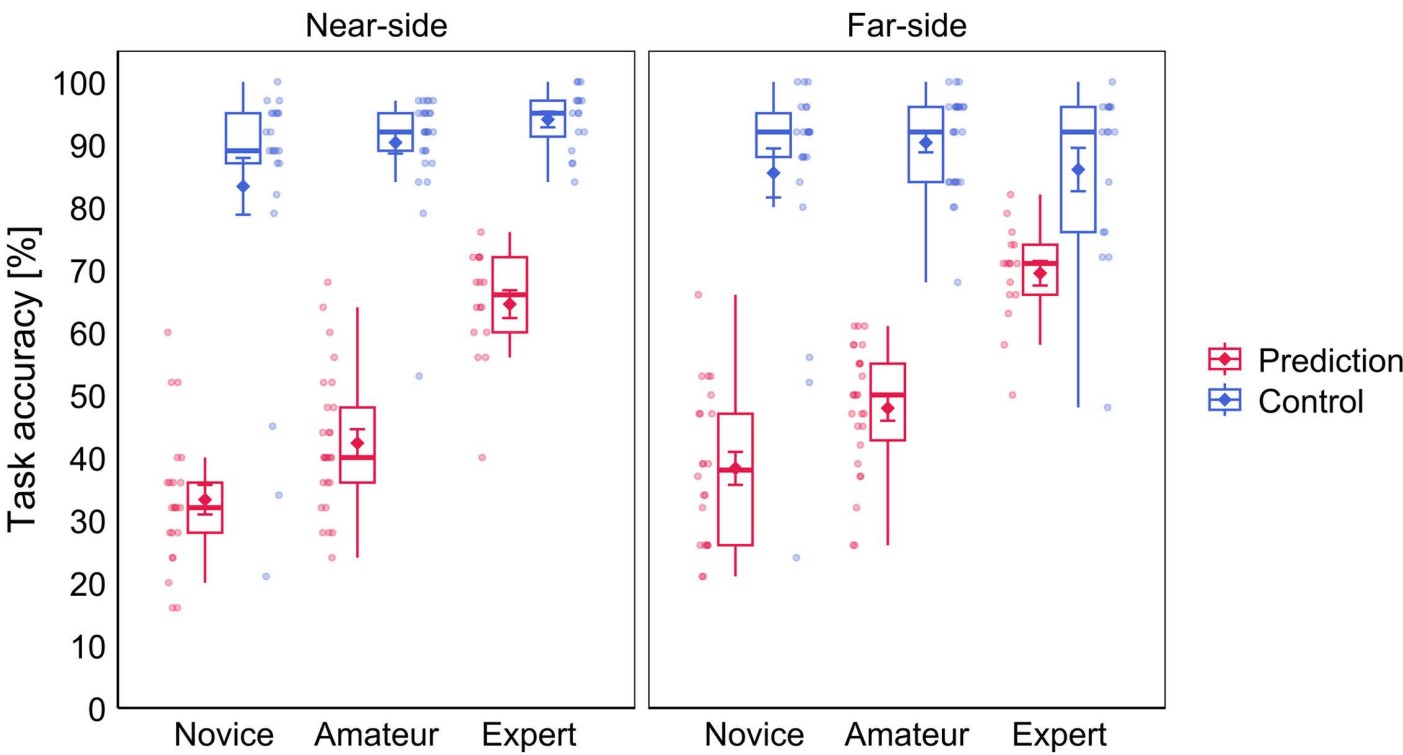

**Fig 3. Performance results for the analysis of accuracy in the tactical decision-making task.** The left and right panels represent the results from the Near-side and Far-side blocks and the Prediction and Control conditions are represented in red and blue, respectively. Jitter dots beside the boxplots represent individual participants of the respective groups Novice, Amateur and Expert. Diamond symbols with error bars represent the mean and standard errors for each condition.

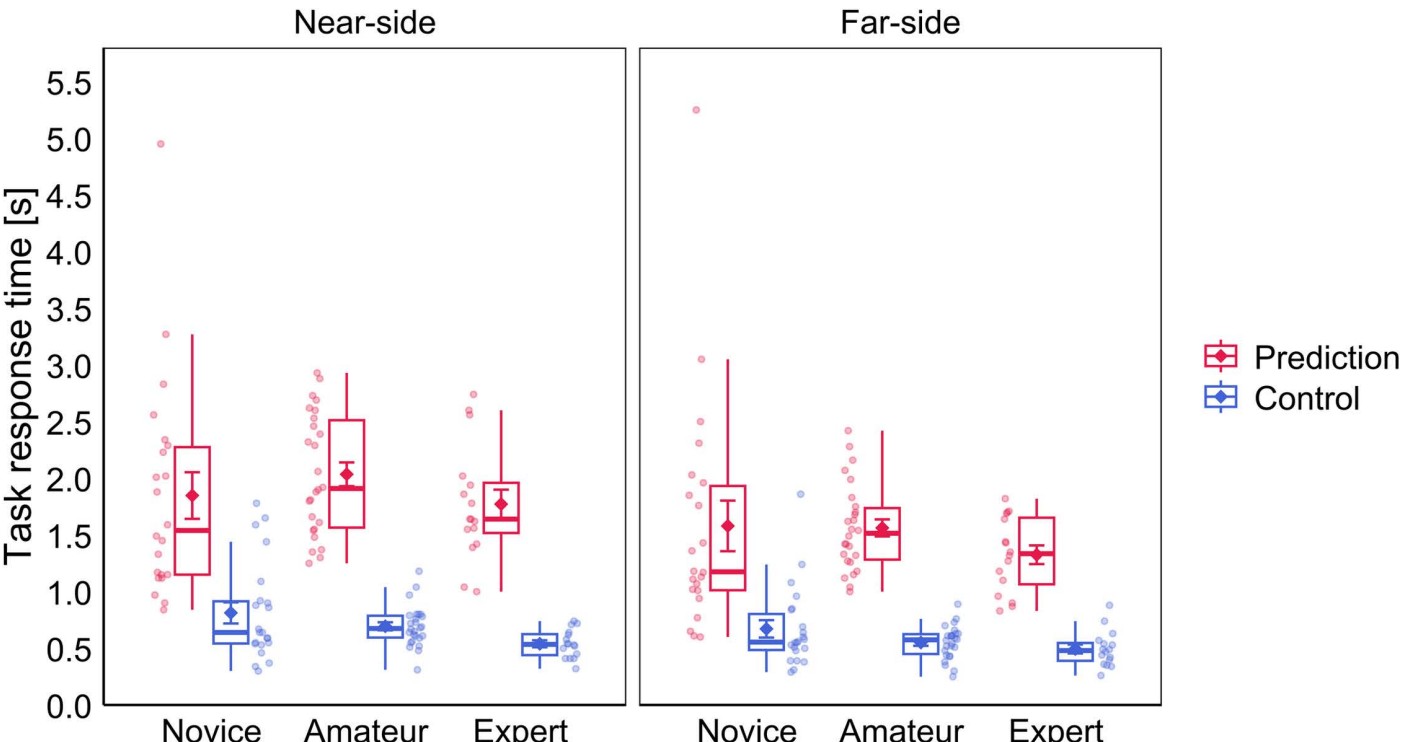

**Fig 4. Performance results for the analysis of response time in the tactical decision-making task.** The left and right panels represent the results from the Near-side and Far-side blocks and the Prediction and Control conditions are represented in red and blue, respectively. Jitter dots beside the boxplots represent individual participants of the respective groups Novice, Amateur and Expert. Diamond symbols with error bars represent the mean and standard errors for each condition.

**Table 2. ANOVA results for task accuracy and response time analyses.**

| | Effect: | df | F | p | ηp2 |
|---|---|---|---|---|---|
| **Task accuracy:** | Group*** | 2, 61 | 20.20 | <.001 | .398 |
| | Court | 1, 61 | 2.71 | .105 | .043 |
| | Task*** | 1, 61 | 521.78 | <.001 | .895 |
| | Group × Court | 2, 61 | 2.30 | .109 | .070 |
| | Group × Task*** | 2, 61 | 19.30 | <.001 | .388 |
| | Court × Task*** | 1, 61 | 13.00 | <.001 | .176 |
| | Group × Court × Task | 2, 61 | 2.03 | .140 | .062 |
| **Task response time:** | Group | 2, 61 | 1.59 | .213 | .049 |
| | Court*** | 1, 61 | 82.20 | <.001 | .574 |
| | Task*** | 1, 61 | 653.34 | <.001 | .915 |
| | Group × Court | 2, 61 | 0.73 | .487 | .023 |
| | Group × Task* | 2, 61 | 4.88 | .011 | .138 |
| | Court × Task* | 1, 61 | 4.77 | .033 | .073 |
| | Group × Court × Task | 2, 61 | 1.73 | .186 | .054 |

Significant ANOVA effects are indicated with asterisks (*: $p < .05$; ***: $p < .001$)

ERD of the theta band. The ANOVA for frontal midline theta ERD/S during the Rally phase yielded significant main effects for the factors Court and Task, as well as a significant interaction effect for Court × Task. No other main or interaction effects were significant. Post-hoc comparisons for the interaction effect Court × Task revealed that stronger theta ERS was observed across all groups in the Near-side ($M$ = 7.46, $SD$ = 22.5) compared to the Far-side ($M$ = 0.01, $SD$ = 20.4) block in the Prediction condition ($t(59)$ = 3.17, $p$ = .002). There was no difference between Near-side ($M$ = -3.99, $SD$ = 19.1) and Far-side ($M$ = -3.24, $SD$ = 20.5) in the Control condition ($t(59)$ = -0.33, $p$ = .745). Furthermore, theta ERS was stronger in the Prediction condition compared to the Control condition during the Near-side block ($t(59)$ = 4.79, $p$ < .001) but not during the Far-side block ($t(59)$ = 1.24, $p$ = .221). The ANOVA for frontal midline theta ERD/S during the Freeze phase yielded significant main effects for the factors Court and Task. No other main or interaction effects were significant. Theta ERS was stronger across all groups during the Near-side ($M$ = 8.63, $SD$ = 23.3) compared to the Far-side ($M$ = -1.92, $SD$ = 22.3) block. Also, theta ERS was stronger during Prediction ($M$ = 6.23, $SD$ = 23.9) compared to Control ($M$ = 0.47, $SD$ = 22.1). See Fig 5 for a visualization of the results for frontal midline theta ERD/S. Complete ANOVA results for the analysis of frontal midline theta ERD/S are presented in Table 3.

Although the ANOVAs revealed no significant group differences, one can see in Fig 5 that novices showed higher frontal midline theta ERS values than amateurs and experts indicating a stronger task-related increase in frontal midline theta power in this group. Post-hoc we calculated one-sample $t$-tests against zero (uncorrected) to see whether groups showed significant frontal midline theta power changes relative to baseline, which are available for inspection in the online materials of this article [49]. These exploratory calculations revealed that only the novices showed significant ERS for Near-side Prediction ($t(20)$ = 2.38, $p$ = .027) during the Rally phase, as well as for Near-side Prediction ($t(20)$ = 2.74, $p$ = .013) and Near-side Control ($t(20)$ = 3.06, $p$ = .006) during the Freeze phase. No other group showed significant task-related power change in frontal midline theta in any of the conditions of both the Rally and Freeze phases (all $p$ ≥ .085).

## Alpha ERD/S analysis

The analyses of alpha ERD/S during the tactical decision-making task are presented separately for the Rally and Freeze phases. Positive values indicate an ERS, negative values indicate an ERD of the alpha band. Note that a stronger alpha ERD is indicated by stronger negative values. The ANOVA for parietal alpha ERD/S during the Rally phase yielded a significant main effect for the factor Court. No other main or interaction effects were significant. Alpha ERD across all participants was stronger during the Far-side ($M$ = -53.9, $SD$ = 27.7) compared to the Near-side ($M$ = -45.8, $SD$ = 30.2) block. The ANOVA for parietal alpha ERD/S during the Freeze phase yielded significant main effects for the factors Court and Task. No other main or interaction effects were significant. Alpha ERD across all participants was stronger during the Far-side ($M$ = -58.6, $SD$ = 25.8) compared to the Near-side ($M$ = -49.5, $SD$ = 28.3) block. Furthermore, alpha ERD was stronger during Prediction ($M$ = -57.9, $SD$ = 27.4) compared to Control ($M$ = -50.2, $SD$ = 27.3). See Fig 6 for a visualization of the results for parietal alpha ERD/S. Complete ANOVA results for the analysis of parietal alpha ERD/S are presented in Table 4.

As with frontal midline theta, the ANOVAs for parietal alpha did not reveal significant group differences. One can see in Fig 6 that parietal alpha ERD was prevalent in all groups, however, ERD appears to be less pronounced in amateurs compared to novices and experts. Here we again calculated post-hoc one-sample $t$-tests against zero (uncorrected) to see whether groups showed

## A  Rally phase - frontal midline theta ERD/S

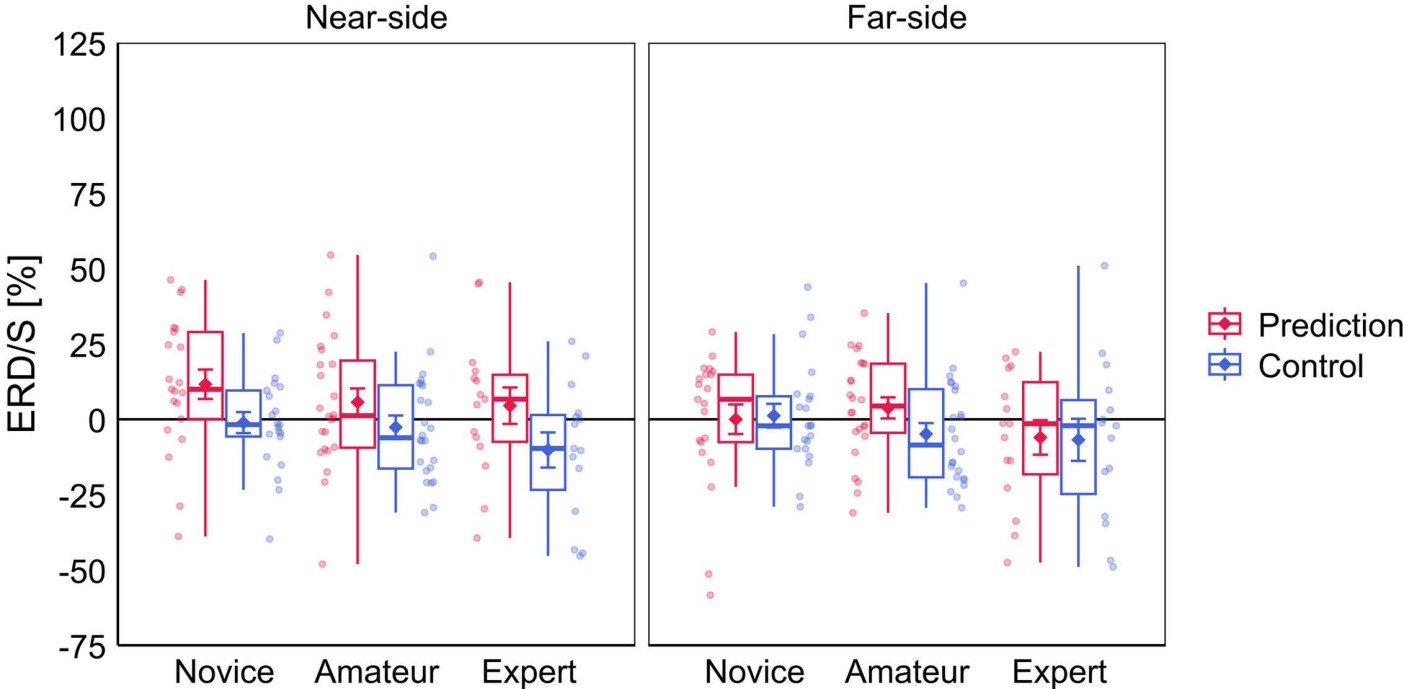

## B  Freeze phase - frontal midline theta ERD/S

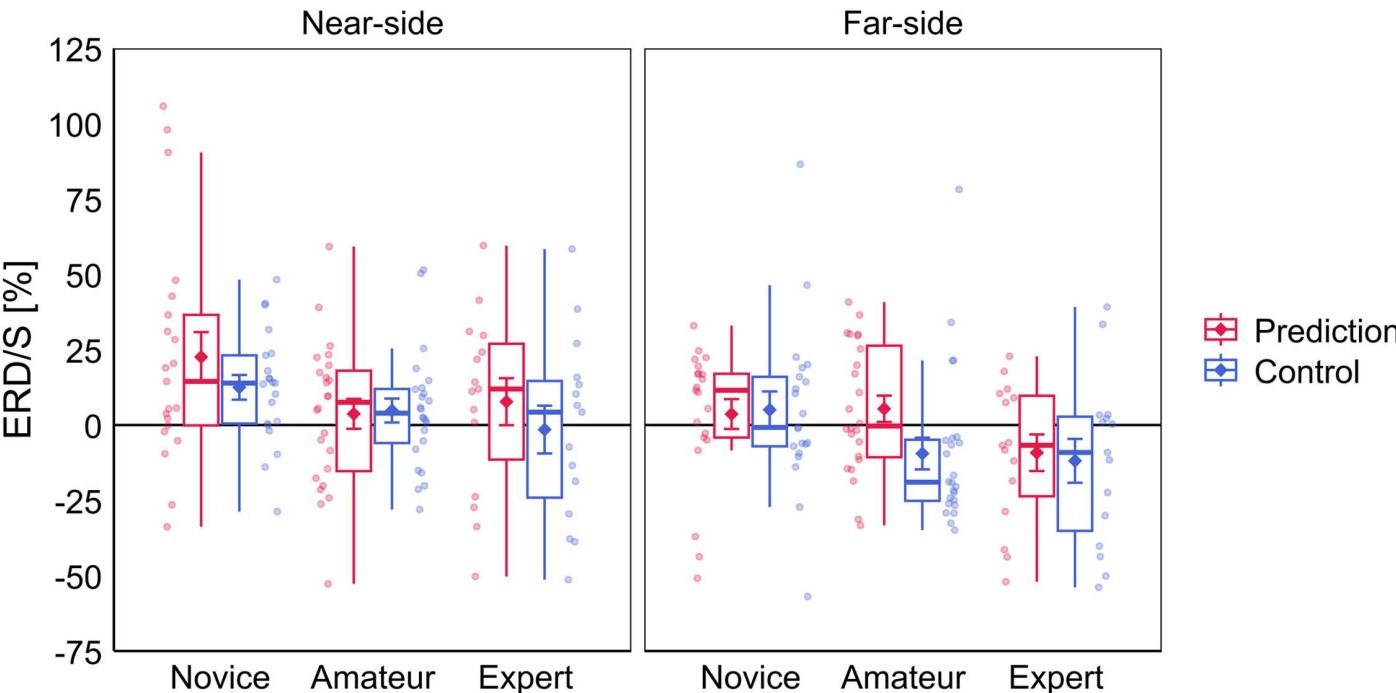

**Fig 5. Results for the analysis of frontal midline theta ERD/S during the tactical decision-making task.** Results for the (A) Rally and (B) Freeze phases of the task are presented, respectively. For each phase the left and right panels represent the results from the Near-side and Far-side blocks and the Prediction and Control conditions are represented in red and blue, respectively. Jitter dots beside the boxplots represent individual participants of the respective groups Novice, Amateur and Expert. Diamond symbols with error bars represent the mean and standard errors for each condition. Note that positive values indicate a theta ERS and negative values a theta ERD.

**Table 3. ANOVA results for frontal midline theta ERD/S analysis.**

| | Effect: | df | F | p | ηp2 |
|---|---|---|---|---|---|
| **Rally phase:** | Group | 2, 57 | 0.85 | .432 | .029 |
| | Court* | 1, 57 | 4.85 | .032 | .078 |
| | Task*** | 1, 57 | 15.69 | <.001 | .216 |
| | Group × Court | 2, 57 | 0.27 | .765 | .009 |
| | Group × Task | 2, 57 | 0.22 | .805 | .008 |
| | Court × Task* | 1, 57 | 6.71 | .012 | .105 |
| | Group × Court × Task | 2, 57 | 2.04 | .139 | .067 |
| **Freeze phase:** | Group | 2, 57 | 2.43 | .097 | .079 |
| | Court*** | 1, 57 | 25.05 | <.001 | .305 |
| | Task* | 1, 57 | 5.72 | .020 | .091 |
| | Group × Court | 2, 57 | 1.29 | .282 | .043 |
| | Group × Task | 2, 57 | 0.11 | .896 | .004 |
| | Court × Task | 1, 57 | 0.02 | .894 | .000 |
| | Group × Court × Task | 2, 57 | 3.10 | .053 | .098 |

Significant ANOVA effects are indicated with asterisks (*: $p < .05$; ***: $p < .001$).

significant parietal alpha power changes relative to baseline, which are available for inspection in the online materials of this article [49]. In the case of parietal alpha, we found significant ERD for all conditions across all groups during both the Rally and Freeze phases (all $p < .001$).

## Discussion

### Behavioral task performance

The present study examined the NEH during a tactical volleyball decision-making task. For this purpose, participants with different levels of prior volleyball experience were divided into three expertise groups, i.e., experts, amateurs, and novices. Participants were asked to view and assess tactical video scenes of volleyball setting situations, while simultaneously their brain activation was recorded using EEG. The task consisted of two conditions, i.e., the Prediction condition and Control condition. During Prediction participants had to judge the outcome of a given setting situation (tactical decision-making), while during Control participants were given the simple task to name the position of the service player during the service. In addition, the task was divided into two blocks, i.e., the Near-side block and Far-side block, in which the videos were presented so that the main actions of interest in each condition occurred either on the near side (court side in front of the net) or on the far side (court side behind the net). We expected differences between the three expertise groups on a behavioral as well as on a neuro-physiological level. On the behavioral level we hypothesized to find higher accuracy as well as shorter response time in relationship to higher expertise. Regarding the aspect of task accuracy our hypotheses were confirmed. We found that novices, amateurs, and experts differed significantly from each other in the accuracy for predicting the outcome of the tactical volleyball setting situations, which in the context of our study is referring to the results from the Prediction condition. Additionally, the accuracy scores of novices, amateurs and experts did not differ for the Control condition. Differences on the behavioral level between expert and novice groups are usually found in the literature [10,29,30]. A similar pattern is found when more than two groups with different expertise level were examined, with behavioral performance measures generally showing a positive relationship with degree of expertise [24,50].

## A  Rally phase - parietal alpha ERD/S

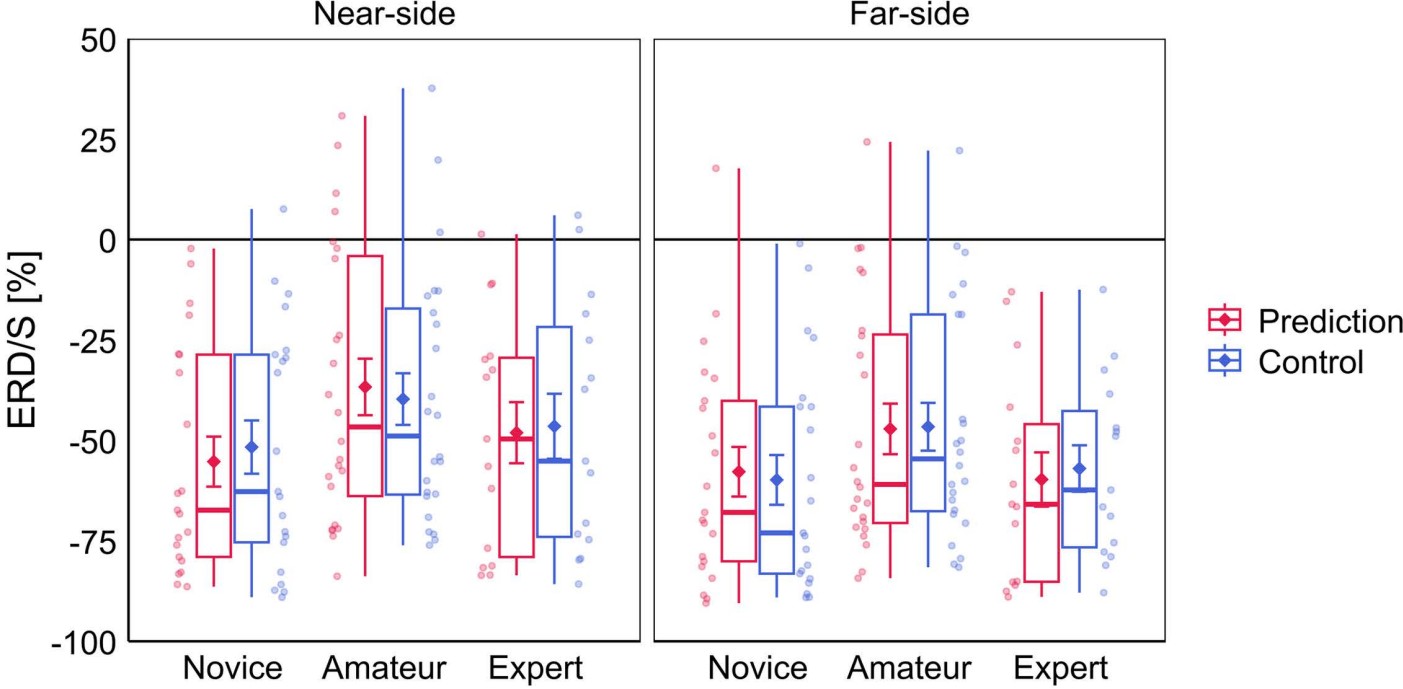

## B  Freeze phase - parietal alpha ERD/S

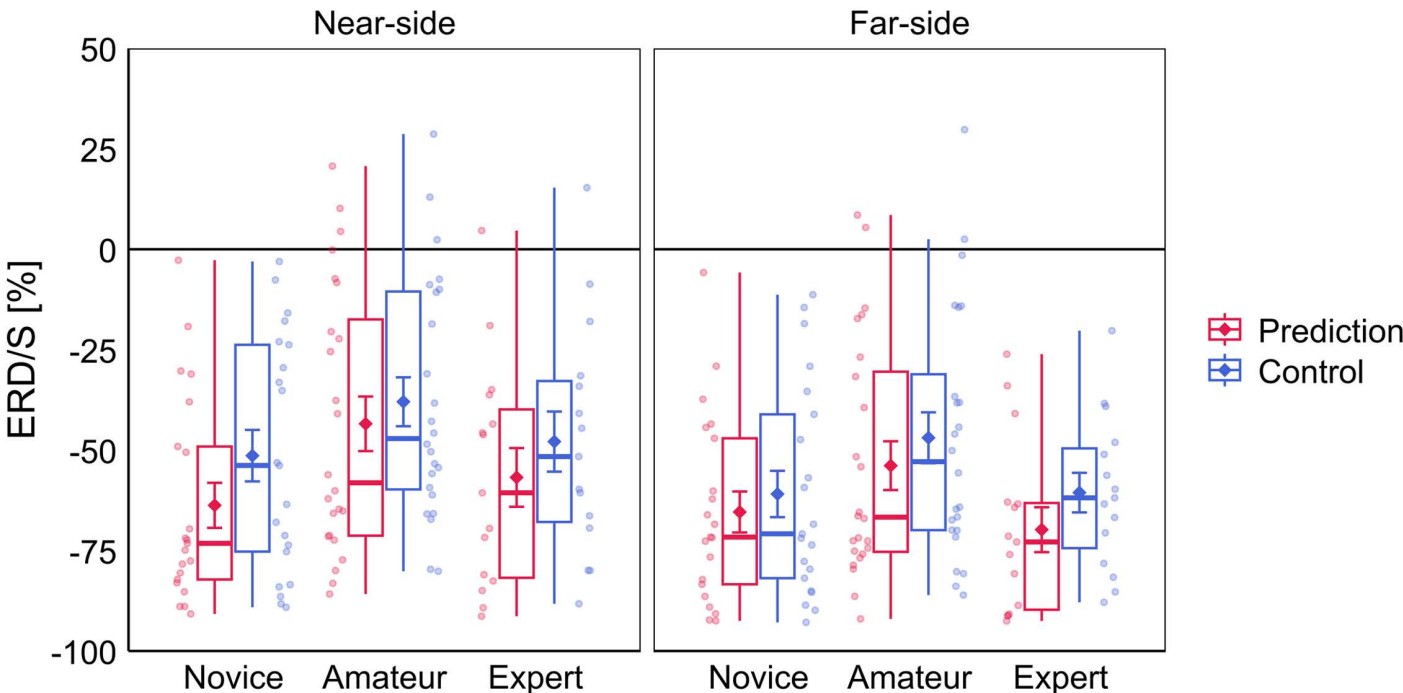

**Fig 6. Results for the analysis of parietal alpha ERD/S during the tactical decision-making task.** Results for the (A) Rally and (B) Freeze phases of the task are presented, respectively. For each phase the left and right panels represent the results from the Near-side and Far-side blocks and the Prediction and Control conditions are represented in red and blue, respectively. Jitter dots beside the boxplots represent individual participants of the respective groups Novice, Amateur and Expert. Diamond symbols with error bars represent the mean and standard errors for each condition. Note that positive values indicate an alpha ERS and negative values an alpha ERD.

**Table 4. ANOVA results for parietal alpha ERD/S analysis.**

| | Effect: | *df* | *F* | *p* | *ηp2* |
|---|---|---|---|---|---|
| **Rally phase:** | Group | 2, 57 | 1.40 | .256 | .047 |
| | Court*** | 1, 57 | 44.47 | <.001 | .438 |
| | Task | 1, 57 | 0.19 | .662 | .003 |
| | Group × Court | 2, 57 | 1.66 | .200 | .055 |
| | Group × Task | 2, 57 | 0.59 | .560 | .020 |
| | Court × Task | 1, 57 | 0.01 | .909 | <.001 |
| | Group × Court × Task | 2, 57 | 1.09 | .343 | .037 |
| **Freeze phase:** | Group | 2, 57 | 2.14 | .127 | .070 |
| | Court*** | 1, 57 | 44.15 | <.001 | .436 |
| | Task*** | 1, 57 | 19.03 | <.001 | .250 |
| | Group × Court | 2, 57 | 2.07 | .135 | .068 |
| | Group × Task | 2, 57 | 0.24 | .784 | .009 |
| | Court × Task | 1, 57 | 0.51 | .480 | .009 |
| | Group × Court × Task | 2, 57 | 1.15 | .323 | .039 |

Significant ANOVA effects are indicated with asterisks (***: $p < .001$).

In the study by Chen et al. [50], participants from three different levels of baseball batting expertise were examined with a baseball-specific anticipation task. Similar to our results they found task performance to increase linearly with level of expertise. They also found a negative relationship between expertise and response time, which contrasts with our results, as we did not find any significant differences in response time between groups. Yet, from visual inspection we see two noticeable patterns in the distribution of the response time data (see Fig 4). First, response times are more spread out in the novices compared to amateurs and experts. Second, we see that on a descriptive level, average response times of novices and experts are quite similar while amateurs show longer response times. We assume most novices responded quicker because they were more guessing rather than giving an informed judgement, given that they had little to no prior volleyball experience. However, some novices possibly tried to invest more effort by contemplating their response longer, possibly explaining the higher variance in this group. Regarding the descriptively longer response times of amateurs compared to experts, we will cautiously suggest that this could be an indicator of the difference in expertise between the two groups. It is a common finding in the literature that skilled athletes show shorter response times compared to less-skilled athletes on a sports-specific task [18]. We assume that what is typically referred to as skilled and less skilled athletes would in our study correspond to experts and amateurs, since those groups are the ones with more substantial prior volleyball experience. Here we see a relation between the speed of responses and the degree of flow experience. On the Flow Short Scale [35] experts scored significantly higher compared to amateurs, which directly refers to their subjective perception of flow experience during the completion of the tactical decision-making task. Although not significant, on average the response times of experts were shorter compared to those of amateurs. The quicker responses of experts could be an indicator of higher fluidity in the completion of the task, which in turn could explain the higher degree of flow. This result is in line with the notion that higher expertise can lead to a higher level of automaticity and fluidity in the performance of a task [17,33,34]. Regarding the Control condition, the response times were consistently shorter compared to the Prediction condition and there were also no significant group differences, which is plausible since the Control condition was designed to be easy and equally difficult for all groups.

Concluding the behavioral performance results, it is necessary to mention that in terms of accuracy and response time we found, that performance was consistently better (higher accuracy, shorter response time) across groups in the Far-side compared to the Near-side block of the task. This could lead to the interpretation that the completion of the task was generally easier when the action of interest was happening on the far side of the court compared to when it was on the near side. However, more plausibly, this pattern resulted from a learning effect, since the Near-side block always came before the Far-side block, giving the participants time to adjust to the demands of the task.

## Theta and alpha ERD/S

Our main interest in this study was to examine whether individuals with higher volleyball expertise would show more efficient brain activation (i.e., neural efficiency) compared to individuals with lower expertise during tactical decision-making. For this purpose, we investigated differences in frontal midline theta and parietal alpha ERD/S between volleyball novices, amateurs, and experts during volleyball-specific tactical decision-making. For the statistical analyses of theta and alpha ERD/S we defined two time periods of interest, namely the Rally phase (last 3 s before the end of each video) and Freeze phase (0.5 s freeze of the last video frame). We expected to find relatively weaker theta ERS as well as weaker alpha ERD in relation to higher expertise. We identified patterns in the theta band that hint towards processes of neural efficiency, yet we also found patterns in the alpha band that conflict with the NEH.

The results from the analyses of the Freeze phase show a coherent picture regarding differences between the two conditions Prediction and Control. During the Freeze phase, relative frontal midline theta ERS was significantly stronger in the Prediction condition compared to the Control condition across all groups. Meanwhile relative parietal alpha ERD was significantly stronger in the Prediction condition compared to the Control condition. Given that increased theta ERS and increased alpha ERD are generally associated with higher cognitive strain [12,14,19–22], these results reflect a successful task operationalization on the neurophysiological level, namely higher cognitive demand in the Prediction compared to the Control condition. Stronger frontal midline theta ERS during Prediction compared to Control was furthermore observed during the Rally phase, however only for the Near-side block of the task.

As our results show, the factor Court also showed quite consistent patterns, albeit the interpretation of this factor is more complex. We found frontal midline theta as well as parietal alpha ERD/S levels to be significantly more positive (i.e., relatively stronger ERS) during the Near-side compared to the Far-side block of the task. As the Near-side block always came before the Far-side block, one explanation for this effect is linked to the order of presentation. Frontal midline theta oscillations which are linked to information encoding processes [19,42,43], were less pronounced in the Far-side block compared to the Near-side block, possibly because the participants were already familiar with the stimulus material when performing the Far-side block. The stimulus material was the same during both blocks (Near-side, Far-side), only the task attached to the individual stimuli (Prediction, Control) changed. Klimesch [19] described hippocampo-cortical feedback loops as the source of frontal theta ERS, which arise during the encoding of new information into memory. We assume that when the participants saw the stimulus material a second time (Far-side block), the amount of new information to extract from the stimuli in order to complete the task was lower, thereby resulting in relatively weaker theta ERS in the frontal midline region compared to the Near-side condition, indicating lower cognitive strain related to information encoding. In general, regarding cognitive strain induced by our task, one must differentiate between the underlying properties of theta and alpha. Generally alpha ERD is associated with stronger cognitive activity [12,14,22]. Hence, the pattern of stronger parietal alpha ERD would indicate higher

cognitive strain during the Far-side compared to the Near-side block. Klimesch [19] furthermore described alpha ERD as the result of thalamo-cortical feedback loops activated by memory retrieval processes. As the participants were already more familiar with the task during the Far-side block, the higher levels of alpha ERD we found possibly reflected stronger memory retrieval processes, as compared to the Near-side block where the stimulus material was still completely new to the participants. We presume the newly encoded knowledge about the task and stimulus material during the Near-side block (reflected by stronger theta ERS), carried over and resulted in stronger memory retrieval processes during the Far-side block (reflected by stronger alpha ERD). However, this interpretation must be taken with a grain of salt. There is also the possibility that these effects arose simply from general temporal effects, related to the order of the stimulus presentation.

Regarding the NEH, our results did not show significant differences in theta and alpha ERD/S between groups, that would confirm our hypotheses. Nevertheless, it seems worthwhile to discuss patterns in the data that could be related to processes of neural efficiency. Firstly, novices showed relatively stronger theta ERS compared to the other two groups (see Fig 5). This was also reflected in the finding that only novices showed significant theta ERS relative to baseline, which is what post-hoc *t*-tests revealed. It is possible, because novices were individuals with little to no prior volleyball experience, that these individuals encoded more new information during the observation of the video scenes, compared to amateurs and experts who were already more familiar with volleyball imagery. This would also be in line with the notion that higher frontal midline theta activity is indicative of information/memory encoding processes [19,42,43]. Secondly, alpha ERD showed a non-linear trend in relation to expertise (see Fig 6). On average, alpha ERD/S levels of novices and experts were rather similar, whereas amateurs showed tendentially weaker alpha ERD, resulting in an inverted U-shaped pattern. When compared to other EEG studies, expert-novice comparisons of task-related alpha power generally show a negative linear relationship between level of expertise and alpha ERD, i.e., weaker alpha ERD in experts compared to novices [23,25–27]. Babiloni et al. [24] furthermore showed this relationship with groups of three different expertise levels in karate during the observation of choreographed karate actions, where novices (referred to as non-athletes) showed the highest, amateurs the second highest and experts (referred to as elite athletes) the lowest levels of alpha ERD over parieto-occipital regions. Clearly our results conflict with that. The inverted U-shaped pattern we found bears more resemblance to the results of Chen et al. [50], which suggested that the amateurs (in their study referred to as intermediate athletes) exhibit relatively stronger brain activation compared to novices, and experts (referred to as skilled athletes) in certain areas of the action observation network, also including the left inferior parietal lobule/sulcus. In contrast, our results suggest amateurs exhibit relatively lower brain activation (weaker parietal alpha ERD) compared to novices and experts. It is difficult to say if and how our results relate to those of Chen et al. [50], also because of the differences in methodology. Nevertheless, given that studies with three or more groups are scarce in the context of the NEH in sport, we wanted to point out this peculiar finding.

To summarize, we consistently found indicators of higher cognitive strain (stronger frontal midline theta ERS, stronger parietal alpha ERD) when participants had to predict the outcome of volleyball setting situations compared to a control condition. We also found that the side of the court where the actions of interest occurred influenced task-related theta as well as alpha power. This was reflected by relatively stronger ERS in frontal midline theta as well as parietal alpha during the Near-side compared to the Far-side block. Moreover, while the distribution of frontal midline theta ERD/S partly resembled the expected pattern (weaker relative ERS in relation to higher expertise) thereby supporting the assumption of higher neural efficiency in higher skilled individuals, the distribution of parietal alpha ERD/S contradicted the expected pattern (weaker relative ERD in relation to higher expertise) and instead showed a non-linear relationship.

## Conclusion

In this study we examined tactical expertise in volleyball on a behavioral and neurophysiological level. A tactical decision-making task, which included the observation and judgement of volleyball setting situations, was presented to participants which were categorized into three groups, based on prior volleyball experience, i.e., novices, amateurs, and experts. We found significant differences in behavioral performance that reflected our categorization of expertise (i.e., higher expertise equals higher performance). On a neurophysiological level significantly stronger theta ERS as well as stronger alpha ERD was observed during active judgment of tactical volleyball situations compared to a control condition, reflecting higher cognitive strain during tactical cognition. On a descriptive level, indications of neural efficiency were identified for the theta band, albeit our analyses did not yield significant group differences. Significant theta ERS in novices, which was not present in amateurs and experts, hinted towards stronger cognitive activity linked to information encoding processes, whereas alpha ERD appeared to follow a non-linear distribution, where amateurs appeared to have more efficient brain functioning compared to novices and experts. With our results we present new insights into the brain functioning of volleyball athletes. Furthermore, we demonstrate the feasibility of differentiating individuals based on their tactical volleyball expertise.

## Supporting information

**S1 Tables. Supplementary tables.** Additional tables showing complementary information and analyses.
(PDF)

## Acknowledgments

The authors would like to thank Mr. Markus Günther for his valuable help in promoting our study to local professional volleyball clubs, as well as Christoph Anzengruber for his technical help with the laboratory setup. This study was supported by the Field of Excellence COLIBRI (Complexity of Life in Basic Research and Innovation, University of Graz).

## Author contributions

**Conceptualization:** Thomas Kanatschnig, Norbert Schrapf, Christof Körner, Markus Tilp, Silvia Erika Kober.

**Data curation:** Thomas Kanatschnig.

**Formal analysis:** Thomas Kanatschnig.

**Investigation:** Thomas Kanatschnig, Lisa Leitner.

**Methodology:** Thomas Kanatschnig.

**Project administration:** Thomas Kanatschnig, Silvia Erika Kober.

**Resources:** Norbert Schrapf.

**Supervision:** Markus Tilp, Silvia Erika Kober.

**Validation:** Thomas Kanatschnig.

**Visualization:** Thomas Kanatschnig.

**Writing – original draft:** Thomas Kanatschnig.

**Writing – review & editing:** Norbert Schrapf, Guilherme Wood, Christof Körner, Markus Tilp, Silvia Erika Kober.

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
