## [Decision Letter · Decision Letter 0]

10 Dec 2024

PONE-D-24-33452EEG theta and alpha oscillations during tactical decision-making: An examination of the neural efficiency hypothesis in volleyballPLOS ONE

Dear Dr. Kanatschnig,

Thank you for submitting your manuscript to PLOS ONE. After careful consideration, we feel that it has merit but does not fully meet PLOS ONE’s publication criteria as it currently stands. Therefore, we invite you to submit a revised version of the manuscript that addresses the points raised during the review process.

**ACADEMIC EDITOR: **

Please do follow the instructions of our expert reviewers to improve you manuscript. 

We look forward to receiving your revised manuscript.

Kind regards,

Prof. Dr. Dragan Hrncic, MD, PhD

Academic Editor

PLOS ONE

Journal Requirements:

Reviewers' comments:

Reviewer's Responses to Questions

**Comments to the Author**

1. Is the manuscript technically sound, and do the data support the conclusions?

Reviewer #1: Yes

2. Has the statistical analysis been performed appropriately and rigorously? 

Reviewer #1: Yes

3. Have the authors made all data underlying the findings in their manuscript fully available?

Reviewer #1: Yes

4. Is the manuscript presented in an intelligible fashion and written in standard English?

Reviewer #1: No

5. Review Comments to the Author

Reviewer #1: This study (EEG theta and alpha oscillations during tactical decision-making: An examination of the neural efficiency hypothesis in volleyball) examined how expertise in volleyball affects brain activity during sport-related decision-making. Participants with different levels of experience (three groups) were asked to predict the outcome of volleyball plays. Researchers measured theta and alpha waves, to assess neural efficiency. The results showed that more experienced players were better at making accurate predictions. Both experienced and less experienced players showed increased brain activity during the task, but there were no significant differences between groups. While these findings suggest that expertise is linked to improved performance, the relationship between expertise and brain efficiency in this specific context is more nuanced than the neural efficiency hypothesis might predict.

Introduction:

The research question is clear and relevant. The authors have effectively outlined the theoretical framework, particularly the neural efficiency hypothesis, and its application to the domain of sport. Minor comment is that in Page 5, line 76, when starting to argument the relationship between theta and alpha power changes and their association with task performance, it would be beneficial to provide a clearer and more detailed explanation, especially for readers who may not be experts in the EEG field.

Method:

The researchers have employed a rigorous and appropriate research design to address the research question. The statistical analyses are robust and the data analysis is well-conducted.

Results:

The results are clearly presented. Minor comment: to further strengthen the interpretation of the results, it would be beneficial to visualize the time-course of theta and alpha power changes using Time-Frequency Representations. This would provide a more detailed understanding of the dynamic changes in brain activity during the task.

Discussion:

The discussion is well-argued but the writing style could be improved by using clearer language and avoiding overly complex terminology.

6. PLOS authors have the option to publish the peer review history of their article (what does this mean?). If published, this will include your full peer review and any attached files.

Reviewer #1: **Yes: **Mohammad Ali Nazari

---

## [Author Response · Author response to Decision Letter 0]

7 Jan 2025

Response to Reviewers

Concerning the revision of the manuscript EEG theta and alpha oscillations during tactical decision-making: An examination of the neural efficiency hypothesis in volleyball (PONE-D-24-33452)

Dear reviewers,

Dear editors of PLOS ONE,

In the name of all my co-authors, I would like to thank you for your thoughtful and constructive feedback for our manuscript. We hereby present to you our responses to all points raised during your reviews and the subsequent changes we made to the manuscript.

Kind regards,

Thomas Kanatschnig

Journal Requirements

PLOS ONE's style requirements were adopted.

This work was financed in its entirety by the University of Graz. Therefore, there is no specific funding information to be provided for this work, as no grant has been awarded for it. We would kindly ask to please let the statement: “The authors acknowledge the financial support by the University of Graz.” to be printed in the funding section when this article is accepted for publication.

The supporting file “S1 File” was given the more descriptive name “S1_Tables” and any in-text citations were matched accordingly. File title and legend were kept as previously provided. The file was converted to PDF.

A search for potential retractions in our reference list using the “Retraction Watch Database” did not yield any results.

Reviewer 1

This study (EEG theta and alpha oscillations during tactical decision-making: An examination of the neural efficiency hypothesis in volleyball) examined how expertise in volleyball affects brain activity during sport-related decision-making. Participants with different levels of experience (three groups) were asked to predict the outcome of volleyball plays. Researchers measured theta and alpha waves, to assess neural efficiency. The results showed that more experienced players were better at making accurate predictions. Both experienced and less experienced players showed increased brain activity during the task, but there were no significant differences between groups. While these findings suggest that expertise is linked to improved performance, the relationship between expertise and brain efficiency in this specific context is more nuanced than the neural efficiency hypothesis might predict.

We would like to thank you for your very constructive feedback on our manuscript. In the following we provide our responses to your comments.

Introduction:

The research question is clear and relevant. The authors have effectively outlined the theoretical framework, particularly the neural efficiency hypothesis, and its application to the domain of sport. Minor comment is that in Page 5, line 76, when starting to argument the relationship between theta and alpha power changes and their association with task performance, it would be beneficial to provide a clearer and more detailed explanation, especially for readers who may not be experts in the EEG field.

We expanded the section concerning the relationship between theta/alpha power and task performance with the aim of making it better comprehensible for readers with less expertise in EEG. Thank you for making us aware of this aspect.

Method:

The researchers have employed a rigorous and appropriate research design to address the research question. The statistical analyses are robust and the data analysis is well-conducted.

Thank you very much for your favorable judgement of our methodology.

Results:

The results are clearly presented. Minor comment: to further strengthen the interpretation of the results, it would be beneficial to visualize the time-course of theta and alpha power changes using Time-Frequency Representations. This would provide a more detailed understanding of the dynamic changes in brain activity during the task.

We greatly appreciate your suggestion concerning the integration of time-frequency representations. Following your advice, we conducted wavelet analyses on the EEG time courses to investigate temporal dynamics underlying our tactical decision-making task. We compared time-frequency patterns between expertise groups (i.e., Novice vs. Amateur vs. Expert) as well as between electrode positions representing our regions of interest (i.e., FCz vs. Pz). While the time-frequency patterns resulting from our analyses show general differences in the distribution of theta and alpha activity between frontal and parietal areas, the groups largely did not differ from each other visually. Also, we could not identify patterns which could meaningfully explain temporal dynamics of task-related theta/alpha activity in the time interval of interest (i.e., -3 until 0.5 s respective to video end). This is why we decided not to include visualizations of this kind in our manuscript, as we thought visual differences between groups were too small to provide meaningful additional information.

Discussion:

The discussion is well-argued but the writing style could be improved by using clearer language and avoiding overly complex terminology.

Thank you for your thoughtful advice. Accordingly, we made changes to the manuscript with the aim of improving understandability and simplifying the wording.

Additional Changes

• Headings were reformatted following the PLOS ONE formatting guidelines.

• Minor formatting changes were performed to figures 1-6.

• Minor formatting changes were performed to table descriptions.

• Minor general changes concerning grammar and formatting were performed to the main text.

• The sections “Data availability”, “Author contributions” and “Competing interests” were removed from the main text.

• The description of the Amateur group under the “Participants” section was refined.

---

## [Decision Letter · Decision Letter 1]

14 Jan 2025

EEG theta and alpha oscillations during tactical decision-making: An examination of the neural efficiency hypothesis in volleyball

PONE-D-24-33452R1

Dear Dr. Kanatschnig,

We’re pleased to inform you that your manuscript has been judged scientifically suitable for publication and will be formally accepted for publication once it meets all outstanding technical requirements.

Kind regards,

Prof. Dr. Dragan Hrncic, MD, PhD

Academic Editor

PLOS ONE

Additional Editor Comments (optional):

Reviewers' comments:

Reviewer's Responses to Questions

**Comments to the Author**

1. If the authors have adequately addressed your comments raised in a previous round of review and you feel that this manuscript is now acceptable for publication, you may indicate that here to bypass the “Comments to the Author” section, enter your conflict of interest statement in the “Confidential to Editor” section, and submit your "Accept" recommendation.

Reviewer #1: (No Response)

2. Is the manuscript technically sound, and do the data support the conclusions?

Reviewer #1: Yes

3. Has the statistical analysis been performed appropriately and rigorously? 

Reviewer #1: Yes

4. Have the authors made all data underlying the findings in their manuscript fully available?

Reviewer #1: Yes

5. Is the manuscript presented in an intelligible fashion and written in standard English?

Reviewer #1: Yes

6. Review Comments to the Author

Reviewer #1: I re-reviewed the revised manuscript and found that the authors have adequately addressed all of my previous comments. I have no further concerns. I recommend the acceptance of the manuscript for publication in PLOS ONE.

7. PLOS authors have the option to publish the peer review history of their article (what does this mean?). If published, this will include your full peer review and any attached files.

Reviewer #1: **Yes: **Mohammad Ali Nazari

---

## [Editor Report · Acceptance letter]

PONE-D-24-33452R1

PLOS ONE

Dear Dr. Kanatschnig,

I'm pleased to inform you that your manuscript has been deemed suitable for publication in PLOS ONE. Congratulations! Your manuscript is now being handed over to our production team.

Kind regards,

on behalf of

Professor Dragan Hrncic

Academic Editor

PLOS ONE